# Compliance of smokeless tobacco supply chain actors and products with tobacco control laws in Bangladesh, India and Pakistan: protocol for a multicentre sequential mixed-methods study

Zohaib Khan [ID],[1,2] Rumana Huque,[3,4] Aziz Sheikh,[5] Anne Readshaw,[6] Jappe Eckhardt,[7] Cath Jackson,[6,8] Mona Kanaan,[6] Romaina Iqbal,[9] Zohaib Akhter,[9] Suneela Garg,[10] Mongjam Meghachandra Singh,[10] Fayaz Ahmad,[2] S M Abdullah [ID],[3,4] Arshad Javaid,[1] Javaid A Khan,[11] Lu Han,[6] Aziz Rahman,[12] Kamran Siddiqi [ID] [6]

For numbered affiliations see end of article.

**Correspondence to**
Dr Anne Readshaw;
anne.readshaw@york.ac.uk

## ABSTRACT

**Introduction** South Asia is home to more than 300 million smokeless tobacco (ST) users. Bangladesh, India and Pakistan as signatories to the Framework Convention for Tobacco Control (FCTC) have developed policies aimed at curbing the use of tobacco. The objective of this study is to assess the compliance of ST point-of-sale (POS) vendors and the supply chain with the articles of the FCTC and specifically with national tobacco control laws. We also aim to assess disparities in compliance with tobacco control laws between ST and smoked tobacco products.

**Methods and analysis** The study will be carried out at two sites each in Bangladesh, India and Pakistan. We will conduct a sequential mixed-methods study with five components: (1) mapping of ST POS, (2) analyses of ST samples packaging, (3) observation, (4) survey interviews of POS and (5) in-depth interviews with wholesale dealers/suppliers/manufacturers of ST. We aim to conduct at least 300 POS survey interviews and observations, and 6–10 in-depth interviews in each of the three countries. Data collection will be done by trained data collectors. The main statistical analysis will report the frequencies and proportions of shops that comply with the FCTC and local tobacco control policies, and provide a 95% CI of these estimates. The qualitative in-depth interview data will be analysed using the framework approach. The findings will be connected, each component informing the focus and/or design of the next component.

**Ethics and dissemination** Ethical approvals for the study have been received from the Health Sciences Research Governance Committee at the University of York, UK. In-country approvals were taken from the National Bioethics Committee in Pakistan, the Bangladesh Medical Research Council and the Indian Medical Research Council. Our results will be disseminated via scientific conferences, peer-reviewed research publications and press releases.

## Strengths and limitations of this study

► This will be the first multicountry study on the smokeless tobacco (ST) supply chain in South Asia that aims to map ST points of sale using geographical information systems.
► Building on the methods used by the global "Tobacco Packs Surveillance System" project, our proposed study will develop/update the inventory of ST products available in Bangladesh, India and Pakistan.
► Purposive sampling of our main study sites may decrease the external validity of our methods and results.
► Owing to a fear of the relevant authorities, the answers to our survey questions on taxes and finances might not be completely accurate.
► We will not be able to do a regression analysis, including shop characteristics, for the pack analysis component, as we are only collecting unique packs and not all packs from each shop.

## INTRODUCTION

Smokeless tobacco (ST) refers to all tobacco products that are used without burning the product.[1] The International Agency for Research on Cancer estimates that ST contains more than 30 carcinogens.[2] The use of ST is associated with an increased incidence of head, neck and gastrointestinal cancers.[3–5] Additionally, ST use is associated with oral potentially malignant disorders[6] and a variety of other oral and systemic diseases.[7 8] An estimated 300 million people around the globe use ST products in some form, and the majority (≥85%) of these ST users live

in South Asia.[1 9] The use of ST products is particularly common in India, Bangladesh and Pakistan, where it is considered an acceptable social norm.[10] This situation is further complicated by the presence of high amounts of tobacco-specific nitrosamines, the main cancer-causing agents in tobacco, in the products available in South Asia, compared with the rest of the world.[11]

The Framework Convention for Tobacco Control (FCTC) is a global treaty that aims to enhance tobacco control measures and incorporate the provisions of the framework into their national tobacco control laws.[12] Although the FCTC is aimed at curbing all types of tobacco use, the focus of tobacco control in most countries has been on controlling smoking, with control of the non-smoking forms of tobacco often neglected.[13 14] The 2016 Global Progress Report on the implementation of FCTC highlighted gaps in the formulation and implementation of ST control policies in most parties to the FCTC.[15] Bangladesh, India and Pakistan are signatories to the FCTC and have made variable progress in implementing the articles of the framework by developing national tobacco control policies.[16]

The tightening up of smoking regulations has been associated with a reported shift from smoking to ST use in some South Asian countries.[9 17] The increase in ST use can be attributed to a variety of factors, including, but not limited to, cultural acceptance, perceived safety of ST products compared with smoking, harm reduction, low prices and lax or absent ST control laws.[14]

A 2018 report from the Global Knowledge Hub on Smokeless Tobacco indicated progress being made with regards to ST control policies and regulations in South Asia.[14] Many of these regulations and laws are aimed at demand reduction via various interventions at the tobacco points of sale (POS), for example, removal of tobacco products' displays, and restrictions and bans on the advertisement of tobacco products at the POS.[18 19] The success of these regulations and laws in curbing the tobacco epidemic in South Asia largely depends on the degree of compliance with the implementation of these laws and regulations.[20] However, the evidence on the implementation of the ST control laws and regulations at the POS in the context of India, Bangladesh and Pakistan is scarce. Most of the previous work carried out in these countries focused on tobacco advertisement and promotion at the POS and health warnings on ST packaging,[21–26] with little emphasis on the other provisions of the FCTC, such as sale to/by minors, price and taxation, and provision of viable alternatives. As such, there is a need for a comprehensive assessment of the compliance of POS in India, Pakistan and Bangladesh with the local ST control policies and the specific articles of the FCTC.

This multicountry sequential mixed-methods study aims to provide a comprehensive assessment of compliance of ST POS and products with the tobacco control laws in our three target countries. Furthermore, we aim to elicit information on any disparities between the implementation of FCTC/local laws for ST and smoking. We will also develop a country-specific inventory of tobacco products, to gain insight into the diversity of the products available in the local market, as well as identifying illicit products. Additionally, we will assess the barriers and facilitators to compliance of the ST wholesale dealers and suppliers with the different provisions of the FCTC/local laws.

## METHODS AND ANALYSIS
### Study design
We will conduct a sequential mixed-methods study, incorporating quantitative work (geomapping of ST POS, in-house analyses of ST products, POS survey interviews and observations) and qualitative work (in-depth-interviews with supply chain actors, eg, wholesale dealers and suppliers). The findings from each component will be connected and used to inform the focus and/or the design of the next component, that is, mapping of the POS will allow us to build a sampling frame from which we will randomly choose survey interview respondents. The ST pack analyses will provide us an insight into manufacturing and sale practices that can be incorporated into the POS survey interview tool and the topic guide for in-depth interviews. The findings from the survey interviews and observations of the POS will guide the development of country-specific topic guides for the in-depth interviews.

### Study sites
We will conduct this study in Bangladesh, India and Pakistan. In each country, we have purposively selected two sites based on the use and diversity of the available ST products. In India, the study will be conducted in the North-East and North-West districts of Delhi. In Bangladesh, the study will be conducted in the districts of Dhaka and Rangpur. In Pakistan, the study will be conducted in the districts of Karachi and Peshawar. In Pakistan and Bangladesh, the selected sites are the hubs of ST use and manufacture; for India, our rationale for choosing the sites was based on Delhi being a metropolitan city with a diverse population and products and also the logistics available to study team.

### Study duration
The data collection will be carried out from 1 June 2019 to 31 December 2020, and analyses will be completed by 30 March 2021.

### General sampling strategy
We will use a multistage mixed sampling technique for the study (figure 1).

### Stage 1
Within each of the selected administrative areas (district/division), we will purposively select one predominantly urban and one predominantly rural subdistrict. Our study sites included districts/areas, which comprise urban, peri-urban and rural areas. We will use the available local government documents, latest census and expert opinion of local researchers to identify two subdistricts/areas at each site,

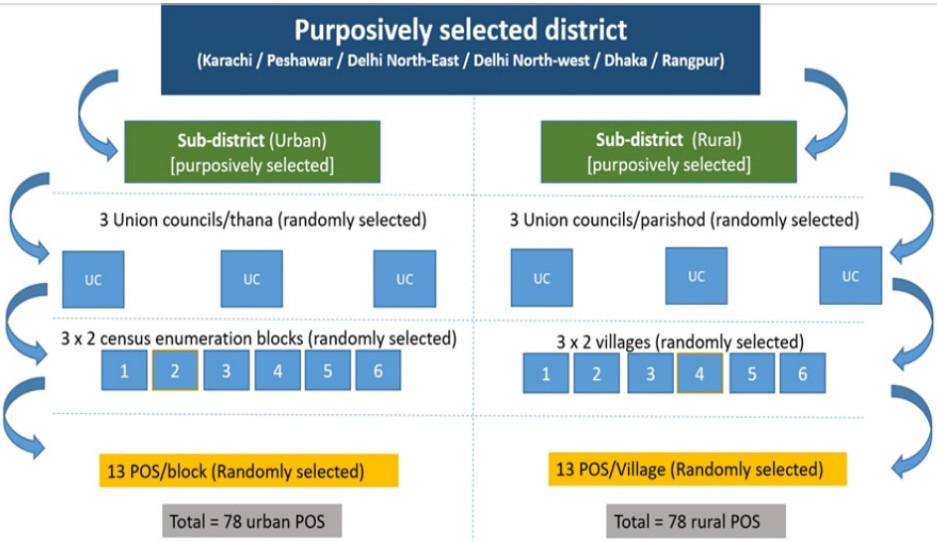

**Figure 1** General sampling strategy to be used at all sites. (a) Within each of the selected administrative areas (district/division), we will purposively select one predominantly urban and one predominantly rural subdistrict. We will use the available local government documents, latest census and expert opinion of local researchers to identify two subdistricts/areas at each site, one predominantly urban and the other predominantly rural/peri-urban. (b) We will randomly select three smaller administrative units per subdistrict. These will be our primary sampling units (PSU). (c) From each PSU, we will randomly select two enumeration blocks/neighbourhood areas/villages; these will be our secondary sampling units (SSU). (d) From each SSU, we will recruit up to 13 points of sale (POS). (e) Required sample size/country is 290; through this strategy, we will be able to recruit up to 312 POS.

one predominantly urban and the other predominantly rural/peri-urban.

## Stage 2
We will randomly select three smaller administrative units per subdistrict. These will be our primary sampling units (PSU).

## Stage 3
From each PSU, we will randomly select two enumeration blocks/neighbourhood areas/villages; these will be our secondary sampling units (SSU).

### Data collection (general considerations)
At each study site, we will deploy a team of trained data collectors to administer the quantitative data collection instruments (comprising the geomapping, in-house analysis of the ST products packaging, structured POS vendor survey interview and observation components). Data collectors will be trained on how to administer the various study tools beforehand. They will also receive training in obtaining informed and written/verbal consent. Calibration for the product in-house analysis component will be done via video conferencing involving study teams from all sites. A qualitative research specialist will train one person at each participating site to conduct and transcribe the in-depth interviews with the suppliers and/or manufacturers. The data will be monitored for quality and completeness by the site leads/study leads/principal investigator. We will also conduct small pilot studies in each country to field-test our study tools.

### Data management
The data will be initially collected on a paper-based questionnaire and then transferred to a central database kept on a secure server at the University of York. Each ST product, POS vendor, supplier and manufacturer recruited in the study will be allotted a unique identification number and the recorded data will not contain any identifiable information about them. Only the study leads/principal investigator/study statistician will have access to the database and data collection tools. All filled questionnaires will be stored in locked filing cabinets at the collaborating institutions in each of the study centres.

In line with the General Data Protection Regulations of the European Union and the Research Governance Framework for Health and Social Care Research, the data will be securely archived by the University of York for a minimum of 5 years after the completion of the study. Following authorisation from the sponsor, arrangements for confidential destruction will then be made. Recordings from the in-depth interviews will be deleted at the end of the study. If an individual withdraws consent for their data to be used, it will be destroyed without breach of confidentiality.

## STUDY-COMPONENT-SPECIFIC METHODS
### POS geomapping
#### Eligibility criteria for POS
A POS will be eligible for inclusion in the study if it sells any form of ST. The common types of tobacco POS that might be eligible for our study are:
1. General/departmental stores.

2. Petrol pump/gas station stores.
3. Beer or liquor stores.
4. Paan (betel quid) shops.
5. Grocery stores
6. Exclusive tobacco shops.
7. Discount shops.
8. Mobile vendors/carts.
9. Stationary carts.
10. Country-specific additional POS types.

### Identifying and mapping eligible POS

To identify the eligible POS, we will employ standard approaches for tobacco POS audits.[27] This will entail:

1. Developing a list of tobacco sellers in the selected areas by collecting the details of local vendors, who have a license to sell tobacco products (if applicable), and/or the acquisition of general lists of local businesses/sellers from trade unions and/or local administrative authorities and combining potential tobacco POS from these lists.
2. This will be followed by 'ground truthing' by trained data collectors, who will visit these shops to ascertain (a) the existence of the POS and (b) the sale of ST products at these POS.
3. Once ascertained, each eligible POS will be allotted a unique identification number and its location will be recorded via the Google Maps digital application. The application gives its users their current location on Google Maps, the Global Positioning System (GPS) coordinates and the geolocation status.
4. If no prior lists (license holders or businesses/sellers) are available at the study location(s), our data collection team will do 'canvassing' of the ST POS by walking along all the primary and secondary roads in that area and identifying eligible ST POS (online supplementary appendix 1).
5. Additionally, we will request each ascertained POS vendor to identify other ST POS, which will be added to the list.
6. If data collectors observe a mobile vendor during the ground truthing or canvassing at the study sites, the area where the team establishes the first contact with that vendor will be marked as his/her location.

### Measuring the density of ST POS

We will collect the most recent official population census data of our selected study sites from the relevant department(s) in each country. If such data are not available, we will use unofficial estimates gathered from the local administration. The density of the POS will be calculated per 1000 population at our study sites.

### Measuring the distance of the POS from schools

We will acquire the list of all schools along with their GPS coordinates, situated in the selected SSU from the local education departments. If the data are not available, our team will visit each school physically, drop a Google Maps pin at the school gate and record the coordinates. We will plot all the ST POS and the schools on a digital map to assess if there is compliance with the local laws that ban the sale of tobacco products in the vicinity (distance will vary according to country) of the schools.

### ST products in-house analysis
#### Eligibility criteria

All brands of ST products available in the markets at the study sites will be procured. For a product to be eligible, it must contain tobacco. The collection of the ST products will be based on the brand names and not the generic product, for example, all brands of Gutkha available in the local market will be collected. Loose products that are not sold in packaging will be eligible, but only one generic sample of the product will be procured. The most commonly sold brand/make of cigarettes, cigars, cigarillos and beedi, according to the selected vendor, will be procured.

#### Sample collection and storage

All ST sale shops/vendors identified in the selected SSU by the geomapping exercise will be approached to procure the ST products that are available in the local markets. We will employ an adapted form of the unique packs sampling technique used for the "Tobacco Packs Surveillance System (TPackSS)" samples collection.[28] One of every unique pack of ST (any pack with at least one difference in an exterior feature of the pack including size, brand name, presentation, colours, cellophane, packaging material (ie, hard or soft tin) and the inclusion of a promotional item) available in the store will be purchased. This sampling will begin at a randomly selected eligible vendor. After the purchase at the initial vendor, at each of the remaining eligible vendors, one of every ST pack that was not already purchased at a previous outlet will be purchased. If the selected vendor does not have any new packs (packs that were not purchased at another outlet), the sample collection team will move onto the next eligible vendor. The sample collection will continue until the team has approached the last eligible vendor. The sample collection team will ascertain the retail price of each procured pack from the vendor. They will also ascertain if the prices are inclusive of tax or not. The sample collection team will neither request nor accept a reduced price, in case the vendor offers a discount due to the large quantity being purchased, other than in those cases where discounted products are already being offered to all customers as a part of a promotion. All such general promotions, and individual 'reduced price' offers, will be recorded on a field notes sheet provided to each data collector.

#### Product analyses

All the procured samples will be sorted by product type, and each sample will be allotted a unique identification number marked on the Ziploc bag with indelible ink. The study team at each participating institution will analyse one sample at a time, carefully locking it back in the bag before moving onto the next sample. A physical inventory of all the products will be developed at each site and the data related to the products, for example, unique identification number,

product family, purchase price and date of purchase, will be fed into a digital inventory form (online supplementary appendix 2) adapted from the TPackSS inventory data entry form.

The in-house assessment of the ST products will involve assessment and recording of their size, price, tax disclosure, ingredient lists, health warnings, manufacturer details, etc. Our study tool (online supplementary appendix 3) has been adapted from the Tobacco Advertisement and Promotion Surveys (TAPS) standard tools. We will collect the samples by brand names and assess whether disparities exist between ST product categories and/or different manufacturers. We will also collect data on the country of manufacture and analyse if differences in compliance with the relevant laws can be explained by differences in the country of manufacture and/or 'smuggled/illicit' designation of the product. The definition of illicit products for each participating site will be developed after consulting local stakeholders. We will also analyse the collected cigarette packs, to assess their compliance with the local laws/FCTC. This will also help us in the identification of any disparity in the local laws for ST and those for cigarettes.

### POS survey interviews and observation
#### Sample size
For the POS observation and interviews, we calculated sample sizes for the individual countries based on the prevalence of non-compliance with specific provisions of the local tobacco control laws in the three countries. The point estimates were derived from the published literature, that is, 20% of shops selling tobacco to minors in Pakistan,[29] 19.4% POS non-compliant with Cigarettes and other Tobacco Products Act (COTPA) regulations regarding sizing of the advertisements in India[30] and 20% POS non-compliant with tobacco advertisement and promotion laws in Bangladesh.[31]

Using the formula sample $n=[DEFF*Np(1-p)]/[(d^2/Z^2_{1-\alpha/2}*(N-1)+p*(1-p)]$, and assuming a 20% prevalence of non-compliance, a CI of 95% and an absolute precision of 7%, the sample size for each country came out to be 251. After accounting for a 15% non-response (based on our ST supply chain study in Pakistan and Bangladesh),[32] the sample size was rounded up to 290 per country. All sample size calculations were carried out in OpenEpi software.[33]

#### Sampling strategy
We will follow the general sampling strategy (figure 1). Once a sampling frame of eligible ST POS in each SSU has been developed through the geomapping, a random sample of the 13 POS vendors will be recruited from each SSU.

#### Recruitment
After providing brief information about the study to the person in charge of the POS at the time of the store observation, we will formally ask them to provide written consent to participate in both the POS observation and POS vendor interview components of the study. Those who do not agree

to participate after receiving the study information will not be enrolled in the study. The reasons for declining to participate in the study will be recorded. Following consent, the data collector will proceed to data collection. The consent provider, who could be either the POS owner (not landlords), POS manager or sales assistant(s), will be eligible for an interview.

#### Data collection and tools
An adapted version (online supplementary appendix 4) of the Standardized Tobacco Assessment for Retail Settings tool[34] will be used to record the observations about product promotion, advertisement, health warnings, etc. We will observe both the exterior and interior of the shops to assess compliance with FCTC/local laws. The tool consists of data fields on health warnings inside and outside the shop, product advertisement, promotions, display shelves, dummy packs, product placement, products sold as confectioneries and cross-promotion. We will also collect similar data for cigarettes and other smoking products, for comparison.

Our interview tool (online supplementary appendix 5) has been adapted from a survey instrument that has already been field-tested in the UK, Bangladesh, Nepal and Pakistan.[32] The tool will be translated into local languages at each study site. In addition to a series of close-ended questions on ST products regarding their supply, sale, popularity, prices, profits and taxes, the tool also includes similar questions for cigarettes, to allow for comparative analysis. We will collect data on mediating/moderating variables that could act as barriers/facilitators to the increased implementation of the FCTC/local laws. These variables are awareness of local tobacco control legislation, standards and obligations, awareness of adverse health effects of ST, the effect of health warnings on sales, market share of ST products, overall sales related to the availability of tobacco products for non-exclusive vendors, incentives by manufacturers/suppliers, profit margins, regular inspections, penalties for non-compliance and availability of viable alternatives.

#### Data analysis
The main statistical analysis will report the frequencies and proportions of shops that comply with the WHO FCTC and local tobacco control policies, as measured by the different items on the survey questionnaire, and provide a 95% CI of this estimate. We will also provide summaries for the other categorical variables using frequencies and percentages and for continuous variables, we will provide means and SD in addition to a graphical representation. Medians and IQRs will also be computed if the data are skewed.

We will use logistic regression to explore the bivariate association of compliance with shop characteristics, such as the type of shop, adjusting for the study design. We will further expand these regression models to explore the association of compliance with multiple explanatory variables adjusting for the study design. Where possible, figures related to ST sales will be compared with those of cigarette sales, using similar approaches to the above. We will consider a p-value of less than 0.05 to be statistically significant. A detailed

statistical analysis plan will be developed before the main analysis, which will include any a priori subgroup analyses and strategies for handling missing data. We will use STATA V.15 for analysis.

### In-depth interviews (supply chain)
#### Participants, sampling and recruitment
At each of the six study sites, we will identify, and collect contact details of, at least five supply chain actors (from manufacturers, wholesalers and suppliers) with the help of the POS survey respondents. From this list, we will purposively recruit three to five participants at each site, ensuring that the sample includes at least one manufacturer and one supplier from each site.

#### Data collection
A topic guide will be developed for the interviews. The contents of the guide will be based on the findings from the POS survey. That is, the responses in the survey that warrant further exploration will be identified by the study teams in each country and the topic guide aimed at eliciting further information on the responses will be developed by the qualitative research team. We will start with a standard topic guide and then tailor it to the findings from the different countries.

The interviews will be conducted at locations that are easily accessible to participants, and where they are confident to speak openly, such as their workplaces or the offices of ward/union council office bearers or the village chief's house/hujra. Each interview will last 30–60 min. A trained researcher will conduct the interviews in the local language. All interviews will be audio recorded.

#### Data analysis
The interviews will be transcribed verbatim, translated into English and analysed using the framework approach.[35] The framework approach is not associated with a particular epistemological viewpoint or theoretical approach and, as such, is not necessarily concerned with generating social theory. The approach can be used in inductive or deductive analysis or a combination of the two.[36] We will be using an inductive approach and develop a framework matrix, informed by the study objectives, the interview topics and data from preliminary interviews. The data from the three countries will be charted into the same matrix; and then reviewed and interrogated to compare and contrast views, seek patterns, connections and explanations within the data, including comparison across the three countries. Excel software will aid data handling.

#### Patient and public involvement
The study is supported by stakeholder groups in each participating country. The stakeholder groups consist of health professionals, policymakers, representatives from general society, students and influencers. The stakeholder groups meet once every year, and so far have been involved in the design of the study tools, refinement of research questions and data collection methods. These groups will also be involved in the dissemination of findings from the study.

## ETHICS AND DISSEMINATION
We will adhere to the fundamental principles of human rights as per the Declaration of Helsinki. Ethical approval has been obtained from the University of York 'Health Sciences Research Governance Committee'. This work is being carried out by the ASTRA Global Health Research Group (Addressing Smokeless Tobacco use and building Research capacity in south Asia). Each participating South Asian institution in ASTRA has obtained ethical approvals from their institutional/national ethics bodies (online supplementary appendix 6). All potential participants will be provided with information regarding the project and written/verbal consent will be taken from each participant before recruitment. Consent forms will contain personal details, including name, date of birth and address.

All information collected during the study will be kept strictly confidential. There will be restricted access and disposal arrangements for the participants' details. Information will be held securely on paper/electronically at the University of York. The data (other than consent forms) transferred to or from the University of York will be coded with a study enrolment number and will not have participants' identifiers. We have studied the conditions under which disclosures to law enforcement agencies are required and are satisfied that the chances of these occurring (criminal activities) are highly unlikely.

To engage our academic audience and the research fraternity, we will publish the results of the study in peer-reviewed journals. We will also present our findings at an international public health conference and an international tobacco control conference. We will also publish an abstract of our findings on the websites of all the partner institutions. We will also present our work to regional research meetings. Additionally, we will incorporate the learning from this project into academic lectures for master's students of public health. We have already established national stakeholder groups in all three target countries, who will guide the study conduct and help us in the dissemination of our findings.

#### Author affiliations
[1]Office of Research, Innovation, and Commercialization, Khyber Medical University, Peshawar, Pakistan
[2]Institute of Public Health and Social Sciences, Khyber Medical University, Peshawar, Pakistan
[3]Department of Economics, University of Dhaka, Dhaka, Bangladesh
[4]Department of Research and Development, ARK Foundation, Dhaka, Bangladesh
[5]Division of Community Health Sciences, University of Edinburgh, Edinburgh, UK
[6]Department of Health Sciences, University of York, York, UK
[7]Department of Politics, University of York, York, UK
[8]Vaild Research Ltd, Wetherby, UK
[9]Department of Community Health Sciences and Medicine, Aga Khan University, Karachi, Pakistan
[10]Department of Community Medicine, Maulana Azad Medical College, New Delhi, Delhi, India
[11]Department of Pulmonary medicine, Aga Khan University, Karachi, Pakistan
[12]Faculty of Health, Federation University Australia, Ballarat, Victoria, Australia

**Contributors** ZK, KS, RH, SG and MMS conceived the study and wrote the first and all subsequent drafts. AReadshaw, FA, ARahman, ZA, SMA, RI, AJ, JAK and AS wrote the initial drafts and commented on the final draft. MK and LH commented on all

drafts and wrote up the sampling and analyses section. JE and CJ commented on initial and final versions of the draft and wrote up the section on mixed methods and in-depth interviews.

**Funding** This research is funded by the UK's National Institute for Health Research (NIHR) [ASTRA (Grant Reference Number 17/63/76)]. The views expressed in this publication are those of the authors and not necessarily of the NIHR or the Department of Health and Social Care.

**Competing interests** None declared.

**Patient and public involvement** Patients and/or the public were involved in the design, or conduct, or reporting, or dissemination plans of this research. Refer to the Methods section for further details.

**Patient consent for publication** Not required.

**Provenance and peer review** Not commissioned; externally peer reviewed.

**ORCID iDs**
Zohaib Khan http://orcid.org/0000-0002-1885-8254
S M Abdullah http://orcid.org/0000-0003-2083-2253
Kamran Siddiqi http://orcid.org/0000-0003-1529-7778

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
