## [Reviewer comments · BMJ Open]

ARTICLE DETAILS

TITLE (PROVISIONAL)	Compliance of smokeless tobacco supply chain actors and products with Tobacco Control laws in Bangladesh, India and Pakistan: Protocol for a multi-centre sequential mixed-methods study.
AUTHORS	Khan, Zohaib; Huque, Rumana; Sheikh, Aziz; Readshaw, Anne; Eckhardt, Jappe; Jackson, Cath; Kanaan, Mona; Iqbal, Romaina; Akhter, Zohaib; Garg, Suneela; Singh, Mongjam; Ahmad, Fayaz; Abdullah, S M; javaid, arshad; A.Khan, Javaid; Han, Lu; Rahman, Aziz; Siddiqi, Kamran

VERSION 1 – REVIEW

REVIEWER	Marita Hefler Menzies School of Health Research, Darwin, Australia
REVIEW RETURNED	10-Feb-2020

GENERAL COMMENTS	Given tobacco law implementation and enforcement can be inadequate in LMICs, and therefore undermine the effectiveness of FCTC provisions, monitoring of compliance is an important area of research. This study will provide up-to-date data about compliance with smokeless tobacco product health warnings in Bangladesh, India and Pakistan, all countries with high prevalence of ST use, and correspondingly high morbidity and mortality burdens. It is noted that the study is already underway (data collection from 1 June 2019 to 30 May 2020). This review provides comments related to the need for greater clarity and suggestions that are possible within the study timeframe. Title: The title doesn't adequately capture the true scope of the study, given that it will not only provide compliance data but a details about the range and availability of products and vendors. Suggest it be revised to reflect this broader scope. Strengths and limitations: this section could be revised. The first point doesn't really say anything; it would be better to focus on the level of detail that will be provided about compliance with laws. It would also be good to note the limited geographic sites as a potential limitation. Introduction: The 4th paragraph in the introduction states that the implementation of ST control laws and regulations in the study countries is scarce. However, there have been a number of studies that have specifically examined ST package compliance in India and Bangladesh that have not been cited:  • Tobacco Pack Surveillance System (TPackSS). Smokeless Tobacco Health Warning Label Compliance: India-2016. Baltimore, MD: Johns Hopkins Bloomberg School of Public Health. https://www.globaltobaccocontrol.org/tpackss/sites/default/files/tpackss_smokeless_HWL_10_22_2018.pdf
--

	 • Iacobelli M, Saraf S, Welding K, et al. Manipulated: graphic health warnings on smokeless tobacco in rural India. Tobacco Control Published Online First: 29 January 2019. doi: 10.1136/tobaccocontrol-2018-054715 • Mullanpudi S, Britton J, Kulkarni MM, Moodie C, Kamath VG, Kamath A. A pilot study to assess compliance and impact of health warnings on tobacco products in the Udupi district of Karnataka State, India. Tobacco Induced Diseases. 2019;17(May):45. doi:10.18332/tid/105894. • Rahman SM, Alam MS, Zubair A, et al. Graphic health warnings on tobacco packets and containers: compliance status in Bangladesh. Tobacco Control 2019;28:261-267. Some studies have also been conducted of cigarette pack compliance, which provide indications of compliance issues generally and add to the rationale for conducting this study:  • Cohen JE, Brown J, Washington C, et al. Do cigarette health warning labels comply with requirements: A 14-country study. Prev Med 2016;93:128–34. doi:10.1016/j.ypmed.2016.10.006 • IGTC. Tobacco pack surveillance system: cigarette health warning label compliance, Bangladesh. 2016 http://globaltobaccocontrol.org/tpackss/sites/default/files/Tpackss_Bangladesh_healthwarningII_090517.pdf In the above paragraph, there is mention of demand reduction “via various interventions at the tobacco points of sale” – it would be good to include examples of these, as not all BMJ Open readers will be familiar with tobacco control strategies. Methods and analysis  • Study design: it would be helpful here to summarise the different components as per the headings (in the same order) as listed in “Study component-specific methods”. As currently written, qualitative interviews with suppliers are mentioned first, followed by the various quantitative methods which makes the following paragraphs somewhat unclear to follow. Also, there is mention of vendor interviews – however, as outlined in the methods, these are really surveys that are administered face-to-face. To avoid confusion with the in-depth qualitative interviews with suppliers, it would be clearer to use the term survey (or questionnaire) in relation to vendors. • Data collection (general considerations): please clarify if all quantitative data collection (observations, purchase of products, vendor surveys) in each store is dependent on obtaining informed and written consent. If so, how will a high rate of refusals be managed to avoid skewing results? There is also no mention in either this section or the general sampling section of the qualitative interviews with suppliers. • Data management: does ‘participants’ in this section refer to ‘store’, ‘vendor’ (who participates in a survey), suppliers who participate in a qualitative interview? Needs to be clearer. There is also mention here of focus group discussions, however there is no focus groups discussions described elsewhere in the manuscript. (The need for clarification re use of the term ‘participants’ applies throughout the manuscript). • In-depth interviews (supply chain): under ‘Data collection’ it is stated that a topic guide will be developed, based on findings from the POS survey. Will this be tailored in each country? Also in this section, it is mentioned that interviews may take place in the workplaces of participants or offices or homes of people of authority. Is there a possibility that these locations might not be ‘neutral’ and may potentially hinder open communication? The
--	--

	sample size in each study site is stated as five: what is the justification for this sample size?  • Data analysis: more information needs to be provided about the deductive content analysis and categorisation matrix that will be used to analyse the in-depth interviews. It would be worth considering inclusion of an inductive approach, in order to identify any unexpected findings that emerge during this stage. Appendix 1 In the section about 'Purchase of products' it is stated that as the data collectors move through shops, only ST products that were not available/purchased at the previous shop will be purchased. However, given the study aims to use logistic regression to explore the bivariate association of compliance with shop characteristics, it would seem logical to purchase items from all shops, to assess if some shops are more or less likely to sell products that are not compliant with eg labelling requirements. The rationale for not doing so should be included in the methods, and noted as a limitation of the study. Appendix 3  • Section 1, packaging and labelling: Items k, l and m: it would make sense to have these items tailored to the legal requirements in each country (eg pictorial health warning covering 85% of the pack in India). It would also be useful to have an item which measures compliance against a sample pack to assess any deviation (eg size, blurriness/clarity of picture warning etc). Item q – flavour and strengths are two different characteristics, with different implications for health messages. It would make sense to have these as separate items. • Section 2, Price and taxation: item a (loosely sold) – presumably need to add this to instructions for data collectors purchasing products to assess how common this is among different vendors. Appendix 4 The section on sales to/by minors is subjective and therefore potentially unreliable. It also seems unnecessary, given it is covered in Q19 in appendix 5.
--	--

REVIEWER	Rizwan Suliankatchi Abdulkader Ministry of Health, Kingdom of Saudi Arabia
REVIEW RETURNED	26-Feb-2020

GENERAL COMMENTS	I congratulate the authors on embarking on a study that addresses one of the pressing issues in the field of tobacco control in the region and as such the significance of the proposal cannot be overemphasized. Here are a few comments for consideration: Study design I understand that the study sites will be chosen purposively but one has to provide some justification for the site selection and the thought process that went behind the selection. Were these sites capitals, or sites of high prevalence of ST use or sites of conveniences or where the authors have previous experience and hence a good understanding of the local dynamics? It will be good idea to select sites with high prevalence of ST use as per the latest GATS. This would ensure a more comprehensive capture of range of ST products. Sites that are major production hubs or wholesale markets are likely to be of use as well. Study duration It is unclear whether data collection has been completed. As per the description, only 3 out of 12 proposed months of the project
---

	are left. If the data collection has been completed, the review of this protocol would be unnecessary. For the qualitative part, no background social theory has been discussed. This is a prerequisite for any mixed methods study. Ideally, a framework of theories and concepts is required before one can begin to do a mixed methods study. Data analysis There is no mention of statistical treatment for the multi-stage nature of the sampling strategy. It is not clear how the ST and smoking products will be compared as there is no mention of it in the data analysis. Also, more clarity is required on how the geospatial data will be analyzed and utilized. It would be a good practice to mention beforehand how the qualitative and quantitative are going to be seen in unison and how they are going to increase our understanding of the research question together.
--	--

REVIEWER	Michael Iacobelli Institute for Global Tobacco Control Johns Hopkins Bloomberg School of Public Health Baltimore, USA
REVIEW RETURNED	20-Mar-2020

GENERAL COMMENTS	Overview: This is a very timely and much needed addition to the body of evidence on smokeless tobacco in South Asia. It will be interesting to learn about the findings that come from this work. With that said, the protocol would benefit from minor edits and considerations detailed below. Page 5, lines 51-56 Please provide some justification for the selection of these areas in Pakistan, India, and Bangladesh for data collection. Page 6, lines 12-13 Please provide a definition and justification for urban and rural districts/divisions. Page 6, lines 51-52 You mention focus group interview recordings, but you don't mention focus groups in the study methods. Please clarify. Page 8, lines 7-13 Some country-level tobacco control laws require that tobacco vendors remain outside of a set radius from schools/educational establishments, for example 100-yards in India. Please detail how your data collection team will assess these radius requirements and will they be from the center of the school grounds, or will you assess a 100-yard buffer around the entire school compound? Additionally, will the data collection team canvas the SSUs for schools not identified from the local education department? Page 8, lines 19-26 Please expand on how you will be identifying and obtaining the ST products in each retailer visited. For example, you mention gutkha. Gutkha has been banned in India for years, and various illicit forms have been found since the banning. However, a popular work-around has been the sale of dual/twin-packs, one spice-only package and a tobacco-only package. Given the importance many
---

	Indian states (Maharashtra and Assam) have placed on spiced-ST sales by implementing bans in some fashion, particularly dual/twin-pack sales, will the data collection team be identifying these product combinations during their data collection activities? These results might be beneficial for policy purposes. Appendix 1, Inclusion criteria Please detail here or in the main protocol document how field staff will identify each of the tobacco POS. Either include an identification definition guide, or provide details or methods on how field staff will accurately and repeatedly classify a POS consistently across the duration of the study. Appendix 2 Suggest adding a data field for price and/or tax amount printed on the package, and/or inclusion of GST for Indian products.
--	---

VERSION 1 – AUTHOR RESPONSE

Reviewer 1	
Title: The title doesn't adequately capture the true scope of the study, given that it will not only provide compliance data but a detail about the range and availability of products and vendors. Suggest it be revised to reflect this broader scope.	Title revised to reflect the scope of the study.
Strengths and limitations: this section could be revised. The first point doesn't really say anything; it would be better to focus on the level of detail that will be provided about compliance with laws. It would also be good to note the limited geographic sites as a potential limitation.	This section is revised based on the suggestions of the Editor and Reviewer.
Introduction: The 4th paragraph in the introduction states that the implementation of ST control laws and regulations in the study countries is scarce. However, there have been a number of studies that have specifically examined ST package compliance in India and Bangladesh that have not been cited	The text has been updated citing the suggested literature. “Most of the previous work carried out in these countries focused on tobacco advertisement and promotion at the POS and health warnings on ST packaging [21-26], with little emphasis on the other provisions of the FCTC, such as sale to/by minors, price and taxation, and provision of viable alternatives”
In the above paragraph, there is mention of demand reduction “via various interventions at the tobacco points of sale” – it would be good to include examples of these, as not all BMJ Open	Suggestion incorporated “Many of these regulations and laws are aimed at demand reduction via various interventions at the tobacco points of sale e.g. removal of tobacco products displays and restrictions and

readers will be familiar with tobacco control strategies	bans on advertisement of tobacco products at the point of sale” [18, 19].
Study design: it would be helpful here to summarise the different components as per the headings (in the same order) as listed in “Study component-specific methods”. As currently written, qualitative interviews with suppliers are mentioned first, followed by the various quantitative methods which makes the following paragraphs somewhat unclear to follow. Also, there is mention of vendor interviews – however, as outlined in the methods, these are really surveys that are administered face-to-face. To avoid confusion with the in-depth qualitative interviews with suppliers, it would be clearer to use the term survey (or questionnaire) in relation to vendors.	The order of the study components has been corrected, clarifying how one component feeds into the next. We have delineated both by using the terms “In-depth Interviews” and “survey interviews”.
Data collection (general considerations): please clarify if all quantitative data collection (observations, purchase of products, vendor surveys) in each store is dependent on obtaining informed and written consent. If so, how will a high rate of refusals be managed to avoid skewing results?	The issue is addressed in the “ethics and dissemination section” i.e. All potential participants will be provided with information regarding the project and written/verbal consent will be taken from each participant prior to recruitment. Consent forms will contain personal details including name, date of birth, and address. During the pilot we discovered that some vendors were not ready to give a written consent but were ready to answer the survey questionnaire. Which prompted us to make changes to the protocol, allowing verbal consent, to be adequate where written consent was not given. The change was notified to the relevant ethics bodies of all the participating countries. We expect this to reduce the refusal rates and skewing of the findings.
There is also no mention in either this section or the general sampling section of the qualitative interviews with suppliers.	The details are given and have been updated under the “In-depth Interviews sections”. We have however, now included a sentence in the “General considerations for data collection” section: “A qualitative research specialist will train one person at each participating site to conduct and transcribe the In-depth Interviews with the suppliers and/or manufacturers”
Data management: does ‘participants’ in this section refer to ‘store’, ‘vendor’ (who participates in a survey), suppliers who	Revised the sentence for clarity:

participate in a qualitative interview? Needs to be clearer. There is also mention here of focus group discussions, however there is no focus groups discussions described elsewhere in the manuscript. (The need for clarification re use of the term 'participants' applies throughout the manuscript).	"Each ST product, POS vendor, supplier, and manufacturer recruited in the study will be allotted a unique identification number and the recorded data will not contain any identifiable information about them". Text now corrected to refer recording of in-depth interviews instead of focus group discussions.
In-depth interviews (supply chain): under 'Data collection' it is stated that a topic guide will be developed, based on findings from the POS survey. Will this be tailored in each country? Also in this section, it is mentioned that interviews may take place in the workplaces of participants or offices or homes of people of authority. Is there a possibility that these locations might not be 'neutral' and may potentially hinder open communication? The sample size in each study site is stated as five: what is the justification for this sample size?	We will start with a standard topic guide and based on findings from individual countries will slightly adapt it to individual settings. This detail has been added. Text edited to reflect that interview participants can choose a location in which they feel confident to speak openly. Made clearer in text now. Based on our previous experience from Bangladesh and Pakistan (Siddiqi et al.2016), the higher up you go the supply chain, the lessor actors you get. In our opinion 3-5 interviews/study site (Total 18-30) might be enough to reach saturation of information. Additionally we propose a purposive sample in order to get maximum variation within our resources.
Data analysis: more information needs to be provided about the deductive content analysis and categorisation matrix that will be used to analyse the in-depth interviews. It would be worth considering inclusion of an inductive approach, in order to identify any unexpected findings that emerge during this stage.	This section has been changed to describe using the framework approach and incorporating deductive and inductive analysis.
In the section about 'Purchase of products' it is stated that as the data collectors move through shops, only ST products that were not available/purchased at the previous shop will be purchased. However, given the study aims to use logistic regression to explore the bivariate association of compliance with shop	For the purchase of ST products we have followed the TPACKSS methods, which necessitates the procurement of "Unique packs".

characteristics, it would seem logical to purchase items from all shops, to assess if some shops are more or less likely to sell products that are not compliant with eg labelling requirements. The rationale for not doing so should be included in the methods, and noted as a limitation of the study.	Therefore, for the pack analysis component, we will not be able to do a regression analysis including shop characteristics. We have noted it as a limitation of the current study. The ASTRA group will however continue to conduct this study component on a regular basis in the future and we will ensure that we use the strategy suggested by the reviewer in the future iterations.
Appendix 3 Section 1, packaging and labelling: Items k, l and m: it would make sense to have these items tailored to the legal requirements in each country (eg pictorial health warning covering 85% of the pack in India). It would also be useful to have an item which measures compliance against a sample pack to assess any deviation (eg size, blurriness/clarity of picture warning etc). Item q – flavour and strengths are two different characteristics, with different implications for health messages. It would make sense to have these as separate items.	The appendix given here is a generalized form, each country will add their country specific requirements to it. For clarity we have incorporated this information with an asterisk into the tool now. Changes made to item “q” which now only asks about flavouring, added item “r” asking about reduced strength
Section 2, Price and taxation: item a (loosely sold) – presumably need to add this to instructions for data collectors purchasing products to assess how common this is among different vendors.	We are already collecting this information through the survey observation tool (See appendix 4, section 12, item c).
Appendix 4 The section on sales to/by minors is subjective and therefore potentially unreliable. It also seems unnecessary, given it is covered in Q19 in appendix 5.	We agree that this section is liable to subjectivity, however we included it just to double check with the responses from the vendors. Our assumption is that the survey interview will take approximately 30 minutes, during which if a data collector sees sale of ST to/or by a minor, he should note it irrespective of the response the vendor gives during the survey. We have trained our data that they should make an effort to confirm the age of the potentially minor buyer/seller for confirmation. This information has been added with an asterisk in the relevant section in appendix 4.

Reviewer 2

I understand that the study sites will be chosen purposively but one has to provide some justification for the site selection and the thought process that went behind the selection. Were these sites capitals, or sites of high prevalence of ST use or sites of conveniences or where the authors have previous experience and hence a good understanding of the local dynamics? It will be good idea to select sites with high prevalence of ST use as per the latest GATS. This would ensure a more comprehensive capture of range of ST products. Sites that are major production hubs or wholesale markets are likely to be of use as well.	In the manuscript methods section, we have mentioned “We will conduct this study in Bangladesh, India and Pakistan. In each country, we have purposively selected two sites based on the use and diversity of the available ST products”. In Pakistan and Bangladesh the selected sites are the hubs of ST use and manufacture, for India we selected Delhi being a metropolitan city with diverse population. In Delhi selected districts also have rural areas, industrial, resettlement and slums with migrant population where expected use of ST products are supposed to be high and hence availability of variety of ST products which is based on previous experience of the researchers. The limited number and location of the study sites reflect the stage we are at within ASTRA. However, we (ASTRA) do plan to extend the study to other sites based on prevalence of ST per GATS and national surveys, and make it nationally representative in the future.
It is unclear whether data collection has been completed. As per the description, only 3 out of 12 proposed months of the project are left. If the data collection has been completed, the review of this protocol would be unnecessary.	At the time of submission data collection for the quantitative components of the study was going on, and has now been completed. For the qualitative component, our data collection (Interviews) have been planned for 3rd quarter of 2020. However, as of writing data entry (quantitative) has stopped for the foreseeable future owing to COVID -19, and it might result in delays re: qualitative data collection.
For the qualitative part, no background social theory has been discussed. This is a prerequisite for any mixed methods study. Ideally, a framework of theories and concepts is required before one can begin to do a mixed methods study.	The key considerations for a mixed methods study is to be clear on the timing, weighting and mixing (Cresswell, Piano-Clark 2017. Designing and Conducting Mixed Methods research). We have added detail on this in the general considerations sub-section in the methods. We have also added some further details on the qualitative part of the study. That is, we explain that we use the framework approach for our analysis, which is not concerned with generating

	social theory, instead it is a method with particular relevance for informing programme/policy development. The approach can be used in inductive or deductive analysis or a combination of the two as we now explain in the data analysis section.
There is no mention of statistical treatment for the multi-stage nature of the sampling strategy. It is not clear how the ST and smoking products will be compared as there is no mention of it in the data analysis. Also, more clarity is required on how the geospatial data will be analyzed and utilized	Given the study essentially employs a cluster sampling technique, we have accounted for the design effect, during sample size estimation, as mentioned in the sample size calculation in the “POS Survey Interview and Observation” sub-section in the Methods. We will also account for the design in our statistical analysis. Please refer to the “Product analyses” sub-section under the ST products In-house analyses section in the study specific methods. Our main aim for collecting Geo-spatial data is to visually analyze the vicinity of ST POS with the schools, colleges etc. This will be carried out in conjunction with another study, under the ASTRA banner in the same sites, that is collecting data on schools. This has been mentioned in the “Measuring the distance of POS from Schools” sub-section.
It would be a good practice to mention beforehand how the qualitative and quantitative are going to be seen in unison and how they are going to increase our understanding of the research question together.	We have now incorporated a sentence on the inter-relatedness of all the components with each other, in the general considerations sub-section in the methods.
Reviewer 3	
Page 5, lines 51-56 Please provide some justification for the selection of these areas in Pakistan, India, and Bangladesh for data collection.	In the manuscript methods section, we have mentioned “We will conduct this study in Bangladesh, India and Pakistan. In each country, we have purposively selected two sites based on the use and diversity of the available ST products”. In Pakistan and Bangladesh the selected sites are the hubs of ST use and manufacture, for

	India, we chose Delhi being a metropolitan city with diverse population. In Delhi selected districts also have rural areas, industrial, resettlement and slums with migrant population where expected use of ST products are supposed to be high and hence availability of variety of ST products which is based on previous experience of the researchers. The limited number and location of the study sites reflect the stage we are at within ASTRA. We (ASTRA) plan to extend the study to other sites based on prevalence of ST per GATS and national surveys, and make it nationally representative in the future.
Page 6, lines 12-13 Please provide a definition and justification for urban and rural districts/divisions.	Our study sites included districts/areas, which comprise urban, peri-urban, and rural areas. We used the available government documents, latest census and expert opinion of local researchers to identify two sub-districts/area at each site, one predominantly urban and the other predominantly rural/peri-urban, to be considered as “urban” and “rural”. Sources consulted: India http://censusindia.gov.in/2011-prov-results/paper2/data_files/India2/1.%20Data%20Highlight.pdf1. Pakistan http://www.pas.gov.pk/uploads/acts/Sindh%20Act%20No.XVII%20of%202012.pdf http://www.kmc.gos.pk/Contents.aspx?id=84

	http://lgkp.gov.pk/neighbourhood-council/ http://lgkp.gov.pk/wp-content/uploads/2015/04/Village-Neighbourhood-Councils-Detatails-Annex-D.pdf Bangladesh https://www.bangladesh.gov.bd/site/view/union-list/%E0%A6%87%E0%A6%89%E0%A6%A8%E0%A6%BF%E0%A6%AF%E0%A6%BC%E0%A6%A8-%E0%A6%B8%E0%A6%AE%E0%A7%82%E0%A6%B9 http://203.112.218.65:8008/WebTestApplication/userfiles/Image/District%20Statistics/Rangpur.pdf
Page 6, lines 51-52 You mention focus group interview recordings, but you don't mention focus groups in the study methods. Please clarify.	It was an oversight that has been corrected. There are no focus groups.
Page 8, lines 7-13 Some country-level tobacco control laws require that tobacco vendors remain outside of a set radius from schools/educational establishments, for example 100-yards in India. Please detail how your data collection team will assess these radius requirements and will they be from the center of the school grounds, or will you assess a 100-yard buffer around the entire school compound? Additionally, will the data collection team canvas the SSUs for schools not identified from the local education department?	Our main aim for collecting Geo-spatial data is to visually analyze the vicinity of ST POS with the schools, colleges via GIS methods. This will be carried out in conjunction with another study, under the ASTRA banner in the same sites, that is collecting data on schools. GIS Coordinates for the schools will either be gotten from local authorities or by study teams dropping a Pin on google Maps at the gate of the school. This has been updated in the "Measuring the distance of POS from Schools" sub-section.

Page 8, lines 19-26 Please expand on how you will be identifying and obtaining the ST products in each retailer visited. For example, you mention gutkha. Gutkha has been banned in India for years, and various illicit forms have been found since the banning. However, a popular work-around has been the sale of dual/twin-packs, one spice-only package and a tobacco-only package. Given the importance many Indian states (Maharashtra and Assam) have placed on spiced-ST sales by implementing bans in some fashion, particularly dual/twin-pack sales, will the data collection team be identifying these product combinations during their data collection activities? These results might be beneficial for policy purposes.	Please refer to Appendix 3, item “P”, we will be collecting dual pack items if available, and analyzing them. Additionally, refer to Appendix 4, section 12, item “h”, we are collecting data on cross-product promotion, including Gutkha.
Appendix 1, Inclusion criteria Please detail here or in the main protocol document how field staff will identify each of the tobacco POS. Either include an identification definition guide, or provide details or methods on how field staff will accurately and repeatedly classify a POS consistently across the duration of the study.	Have updated the appendix as per suggestion: “The data collectors will ascertain the eligibility of an establishment by first inquiring about the availability of ST at that establishment and then proceeding for procurement of ST samples for the “In-house analysis” component of the study. Once they have established the POS as eligible, they will drop a pin on the Google Map as per the method used for “Mapping of POS”.
Appendix 2 Suggest adding a data field for price and/or tax amount printed on the package, and/or inclusion of GST for Indian products.	We have incorporated the suggestions in appendix 2.

VERSION 2 – REVIEW

REVIEWER	Marita Hefler Menzies School of Health Research, Darwin, Australia
REVIEW RETURNED	19-Apr-2020

GENERAL COMMENTS	The authors have provided a comprehensive response to all reviewers and the paper is much improved. I only have a couple of minor suggestions relating to language and one query about study time frame, as per below: Page 3: suggest changing "...put on a back burner" to "neglected" Page 5: study duration. Does the end date of 30 May 2020 for data collection need to be amended given the Covid-19-related delay for qualitative data collection? Pg 11: Suggest changing "A trained researcher will conduct the interviews in the local language and be digitally recorded" to "A trained researcher will conduct the interviews in the local language. Interviews will be audio recorded."
---

REVIEWER	Rizwan Suliankatchi Abdulkader Velammal Medical College and Research Insitute, Community Medicine
REVIEW RETURNED	03-Apr-2020
GENERAL COMMENTS	All my comments have been addressed

VERSION 2 – AUTHOR RESPONSE

We thank you and the reviewers for their positive comments. The comments have helped us make this a much better manuscript. We have now addressed all the observations raised by the reviewers including the "data collection" time lines and the grammar improvement suggestion.

We have now provided the Ethics Committees details as per your request in the supplementary files and cited it in the main text. To native english speakers have now proof read the manuscript and we are confident that there are no more typographical or grammar mistakes.